# Ecological Packaging: Reuse and Recycling of Rosehip Waste to Obtain Biobased Multilayer Starch-Based Material and PLA for Food Trays

**DOI:** 10.3390/foods14111843

**Published:** 2025-05-22

**Authors:** Yuliana Monroy, Florencia Versino, Maria Alejandra García, Sandra Rivero

**Affiliations:** 1Centro de Investigación y Desarrollo en Ciencia y Tecnología de Alimentos (CIDCA-CONICET-CIC), La Plata 1900, Argentina; ymonroy@quimica.unlp.edu.ar (Y.M.); florencia.versino@ing.unlp.edu.ar (F.V.); sandrarivero@biol.unlp.edu.ar (S.R.); 2Facultad de Ciencias Exactas, Universidad Nacional de La Plata, La Plata 1900, Argentina

**Keywords:** rosehip, by-product revalorization, cassava starch, bioadhesive, polylactic acid, circular economy

## Abstract

This study investigates the valorization of agri-food residues by repurposing industrial rosehip oil waste for sustainable food packaging development. Market demands for environmentally friendly alternatives to conventional packaging materials prompted the development of laminated multilayer materials for trays through thermo-compression, using modified cassava starch with citric acid as a compatibilizer. Physicochemical characterization revealed appropriate surface roughness (Rz of 31–64 μm) and controlled water absorption capacities of the composite materials (contact angle of 85–95°), properties critical for food quality preservation and safety. The incorporation of polylactic acid (PLA) films in the laminates significantly enhanced the mechanical performance, increasing the stress resistance by 5 to 10 times, and improved moisture resistance, showing a 78–82% reduction in the materials’ water absorption capacity and an almost 50% decrease in water content and solubility, depending on the processing method. Results indicated that these biocomposite laminates represent a viable alternative to conventional polystyrene foam trays for food packaging. Two distinct multilayer manufacturing processes were comparatively evaluated to optimize production efficiency by reducing the energy consumption and processing time. This research contributes to circular economy principles by transforming agricultural waste into value-added laminated materials with commercial potential.

## 1. Introduction

As the world’s population continues to grow, the food packaging market is constantly evolving to meet new safety and environmental requirements, whilst simultaneously reducing food waste and the environmental impact of single-use packaging [1]. One of the best ways of decreasing food waste is to extend packaged foods’ shelf life without compromising safety [2]. However, such packaging materials require specific mechanical, thermal, barrier, and antimicrobial properties, and their combinations, to ensure food safety and quality [3]. Consequently, numerous and original research and development efforts have been conducted in this regard, ranging from newly sourced and biodegradable materials [1,4,5,6,7] to tailored composite systems [8,9,10,11,12] and functionalization for active and smart food packaging [13,14,15,16,17,18,19,20]. The challenges lie in achieving cost-effective alternatives to commodity plastic packaging that can be easily scalable with current technologies, without hindering their sustainability.

Plastics for food packaging are often combined materials and multilayered systems. These multilayered structures offer comprehensive protection against various harmful substances whilst simultaneously enhancing barrier functionality and delivering materials that possess the necessary mechanical strength and thermal stability [21]. Yet, these multi-material structures are quite difficult to sort, process, and recycle [22], so greener alternatives are being explored. In this regard, fully biodegradable multilayer packaging systems, engineered to be composted at the end of their life, hold the potential to diminish food waste, avoid recycling difficulties, and minimize the environmental footprint [21]. An alternative to produce biodegradable materials with low cost, adequate mechanical and barrier properties, as well as high water resistance, is the use of polylactic acid (PLA) in biobased multilayer packaging systems to enhance the typical low-water resistance of other biodegradable materials. PLA is a well-known hydrophobic, aliphatic, biobased, and biodegradable commercial polyester with a high molecular weight (>100 kDa) [23].

A promising line of research has focused on designing eco-compatible starch-based matrices and fibrous fillers from agricultural and food wastes, such as sugarcane bagasse, corn husks, yerba mate powder, and malt bagasse, among others [21,24,25,26]. The use of lignocellulosic agroindustrial by-products or wastes presents an interesting strategy to reduce the growing pollution problem while increasing the added value of these resources, leading to substantial benefits for the environment in terms of the overall carbon footprint and waste reduction. Therefore, this study proposes the use of waste resulting from industrial essential oil extraction from the fruit of *Rosa rubiginosa* L. (called *rosehip* in English and *escaramujo* in Spanish).

Commonly known as *sweet brier*, or *rosa mosqueta, Rosa rubiginosa* is a native plant to Asia, almost all of Europe, North America, and Northern Africa, and an exotic and invasive species in Australia, New Zealand, and South America [27,28]. Rosehip has widely expanded throughout the Argentine Patagonia region, especially colonizing areas degraded by fire, forest felling, or livestock grazing [27]. Since 2021, it has been listed by the Ministry of Environment and Sustainable Development in Argentina as an invasive exotic species of controlled use for productive or economic purposes [29]. Although rosehip production information is scarce, the latest reported data collection from 2005 showed that more than half of the global sweet brier production came from Chile and Argentina [30]. The residues resulting from the cold-press processing of rosehip, known as “fluff”, are useless for any commercial purpose and are typically disposed of or used as fertilizer and briquettes sold as heating fuel [31]. Thus, it is of interest to look for alternatives for the valorization of the generated biowaste to achieve a zero-waste production [32]. Torres-Scaincalepore et al. [28,32] studied the use of rosehip oil wastes from a biorefinery approach to extract biopolymers and produce biofuels. Alternatively, this work studies the use of these residues as raw material in eco-compatible composite materials.

Furthermore, eco-friendly adhesives obtained from biomass as a substitute for synthetic adhesives represents a little-explored strategy in the design of food packaging materials. Starch, being a low-cost, renewable, and biodegradable biopolymer, has raised the scientific community’s attention as a replacement for conventional plastics in single-use applications [10]. In particular, cassava starch has been employed as a bonding agent in numerous products owing to its affordability, widespread availability, non-toxic nature, and biodegradability [25,33,34]. Yet, native starch-based adhesives tend to exhibit insufficient bonding strength and limited water resistance, requiring chemical modifications to improve the techno-functional properties of the adhesives [24]. Carboxylic acids, also non-toxic and affordable, have proven to be effective cross-linkers [25,34]. In particular, citric acid (CA) is a natural and inexpensive cross-linker consisting of three carboxylic groups used for converting hydroxyl-containing polymers into reactive functional polymers stabilized by supramolecular interactions or covalent bonds [24,35,36]. These advantages make it a promising candidate for the development of biobased adhesives for food packaging applications [37].

This work focuses on designing fully biobased, biodegradable multilayer food trays through thermo-compression, using rosehip processing waste as a reinforcing filler, modified starch as a natural binder, and a PLA film as the outer layer to enhance mechanical strength and moisture resistance. The thermo-compression process is a key step in the production of composite particulate materials [38]. Therefore, thermo-compression processing conditions were systematically optimized by investigating various processing strategies and their effect on physicochemical, water resistance, mechanical, and microstructural properties of the resultant materials. Significant enhancement in material performance was achieved through comparative analysis of single-step versus two-step processing methodologies, with particular emphasis on the integration and lamination techniques for PLA films. The strategic application of these films to the composite substrate yielded multilayer semi-rigid materials with demonstrably superior moisture barrier capabilities and enhanced mechanical resilience.

This study aims to develop fully biobased, biodegradable composites using rosehip waste as a reinforcing filler and modified starch as a binder, with a PLA film for moisture resistance. The formulation enhances mechanical properties while ensuring eco-friendly packaging performance

The innovative aspects of this research encompass both material formulation and processing methodology, representing a significant advancement in sustainable packaging technology. The novel integration of agricultural waste valorization with biopolymer application presents a commercially viable alternative to conventional petroleum-derived packaging materials at competitive production costs. This approach simultaneously addresses multiple environmental challenges by reducing agricultural waste streams, minimizing dependence on synthetic polymers, and introducing fully biodegradable packaging options for dry and semi-moist food products. The resulting composite materials demonstrate the feasibility of maintaining functional performance requirements while adhering to circular economy principles, potentially facilitating regulatory compliance with emerging restrictions on single-use plastics in food packaging applications.

## 2. Materials and Methods

### 2.1. Materials

The studies were conducted using the industrial residue of rosehip (*Rosa rubiginosa Linnaeus*) essential oils extraction provided by the company Rosa Patagónica S.A. (Río Negro, Argentina), cassava starch (*Manihot esculenta*) from Cooperativa Montecarlo (Montecarlo, Misiones, Argentina), and citric acid (CA; BIOPACK, Buenos Aires, Argentina). Polylactic acid (PLA) grade 4043D in pellet form, containing 98% L-lactide and approximately 2% D-isomer, was used in this study. This particular grade of PLA is intended for the production of films and was purchased from Natureworks LLC (Minneapolis, MN, USA) under the trademark Ingeo.

### 2.2. Characterization of Rosehip Fluff Residue

The chemical composition of the rosehip oil extraction residue (R) was evaluated by determining its cellulose, hemicellulose, and lignin content, using standard methods of acid and neutral detergent fiber (ADF and NDF), and lignin by the Klason method, as well as ash (muffle calcination), lipid (Soxhlet extraction), protein (Kjeldahl method), and moisture (gravimetric method) content according to AOAC [39] standard methods. All results were expressed as % *w*/*w* of the samples on a wet basis.

### 2.3. Elaboration of Sustainable Materials

Citric-acid-modified cassava-starch-based adhesives were formulated following the procedure described by Monroy et al. [24,35] to produce biobased adhesives. Starch-adhesive suspensions, 5% *w*/*w*, were gelatinized at 90 °C for 20 min in a thermostatic bath. Analytical-grade CA was then added to the gelatinized suspensions at concentrations of 30% (weight of polycarboxylic acid/100 g starch). To improve the performance of the formulated adhesives, native cassava starch was added as a filler (5 g native starch/100 g gelatinized starch suspension) at room temperature.

Mixtures of bioadhesive and rosehip residue (particle size < 500 μm) were prepared, and thermo-compression was performed according to a standardized procedure [24,35,40] (Figure 1). Based on previous studies, the R by-product:adhesive ratio was set at 1:1 on a wet basis, resulting in a final content of 90% of filler [24,25,35]. The influence of processing conditions (120 and 130 °C for 3 or 5 min) at a pressure of 300 kg/cm^2^ was studied by evaluating the quality attributes and final properties of the materials, named as laminates (L) from here on.

### 2.4. Design of Multilayer Trays

In order to improve the mechanical properties and decrease water affinity of the biobased composite materials, the development of multilayered materials was proposed. A two-step process was used to prepare the thermo-compressed trays. Firstly, approximately 5 g of PLA was placed on a Teflon liner, placed between the press plates, and thermo-pressed at 150 °C for 3 min under 300 kg/cm^2^. The PLA films were easily peeled from the layers after cooling in the air to room temperature. A similar protocol was followed by Rhim et al. [41]. Secondly, two different processing steps were proposed, and the optimal conditions for each alternative were studied: placement inside the Teflon liner of PLA film+laminate (L)+PLA film for subsequent thermo-compression (130 °C, 3 min; see Figure 1), and PLA film+composite paste (CP)+PLA film and thermo-compression in the optimized conditions mentioned above, as shown in Figure 1. Here, the laminates are the biobased composite materials with rosehip residue particles and starch-based adhesive previously obtained and characterized in Section 2.3, and the adhesive composite paste is the same formulation that has not yet been thermally compressed. The resulting multilayered materials contained 74% of rosehip residue, 8% of starch bioadhesive, and 18% of PLA on a dry weight basis of the material.

### 2.5. Physicochemical Properties of the Materials

#### 2.5.1. Color

The surface color of the materials was determined using a Minolta CR 400 Series colorimeter (Osaka, Japan). The CIELab scale was used, and the luminosity was measured, which describes 100 for white and 0 for black, and the parameters a* (red–green) and b* (yellow–blue) describe the chromaticity coordinates. At least 12 determinations were made for each sample in different randomly located positions. Color differences (ΔE) were calculated with respect to standard plate parameters. A minimum of 12 measurements were taken for each sample during the analysis process, at various randomly selected locations.

#### 2.5.2. Surface Roughness

Mean roughness (Ra) and mean peak-to-valley height (Rz) of the samples were determined using PCE-RT 1200 equipment (Schwyz, Switzerland). At least 15 measurements were taken from both surfaces of each material.

#### 2.5.3. Thickness

Thickness measurement of the materials was performed with a CM-8222 digital gauge (SOLTEC, Florida Oeste, Argentina). Values corresponded to the mean of at least 10 randomly selected measurements.

#### 2.5.4. Density

The material’s density (g/cm^3^) was determined gravimetrically, in accordance with the methodology outlined by Monroy et al. [24].

#### 2.5.5. Water Content and Surface Wettability

The moisture content of the materials was determined upon drying in an oven (GRX 9203A BLUEPARD Instruments Co., Shangai, China) at 105 ± 1 °C until reaching constant weight (dry sample weight). Samples were analyzed at least in triplicate, and results were expressed as g water/100 g material. To study the wettability of the materials’ surfaces, the contact angle was determined using a goniometer Ramé-hart Model 190 (Ramé-hart Instrument Co., Succasunna, NJ, USA). At least 8 determinations were conducted for each sample.

#### 2.5.6. Water Absorption Capacity and Solubility

Samples of 2 by 2 cm were cut and completely immersed in water for 1 and 60 min, and the water excess was removed for weighing. The water absorption capacity (WAC) was estimated as the amount of water adsorbed (weight difference) and expressed as g water/g sample [42]. Then, samples were dried in an oven (GRX 9203A BLUEPARD Instruments Co., Shangai, China) at 105 ± 1 °C until constant weight. The solubility percentage of the samples was calculated according to Rivero et al. [43].

### 2.6. Fourier Transform Infrared Spectroscopy (FTIR)

The infrared absorption spectra of the laminate surfaces were determined by FTIR on a Nicolet iS10 Thermo Scientific (Madison, WI, USA). Spectral data for intensity ratios were performed in attenuated total reflection (ATR) mode over eight repetitions for each treatment. Data were recorded at a resolution of 4 cm^−1^ between wavelengths of 4000 and 400 cm^−1^ for 32 scans.

### 2.7. Scanning Electron Microscopy (SEM)

The surface and cross-sectional microstructures of the specimens were examined by scanning electron microscopy (SEM) using a FEI Quanta 200 scanning electron microscope (Eindhoven, The Netherlands). Analogously to the characterization methods previously described for rosehip-based materials, mechanical properties were analyzed according to the protocol described in Section 2.8 and surface wettability of the material according to the procedure described in Section 2.5.5.

### 2.8. Mechanical Properties

The ability of a package to safeguard food from physical harm relies on its capacity to endure impact-induced breakage during handling. To examine the mechanical properties of the materials, a puncture test was performed using a TAXT2i Texture Analyzer (Stable Micro Systems Ltd., Godalming, Surrey, UK), where at least 10 probes were assayed. A 2 mm-diameter aluminum SMSP/2 probe was utilized for this purpose. The mechanical profiles show the maximum force (N) and distance (mm) that a material can withstand before puncture. The maximum stress (MPa) before break was calculated following standard procedures [35,44].

### 2.9. Statistical Analysis

A two-way factorial design was used to analyze the effect of hot-pressing conditions (time and temperature) on the material properties. The analysis of variance was performed using the InfoStat Software (2012). Differences were determined by Fisher’s least significant difference (LSD) test using a significance level α = 0.05.

## 3. Results and Discussion

### 3.1. Composition of Rosehip Oil Extraction Residue

The composition of the residue is shown in Table 1. It presented a high fiber content (63.68 ± 2.5%) with a high proportion of lignin (approximately 44.3% of the total fiber content) and low protein, lipid, and inorganic matter contents. Moisture was lower than 10%, in line with other previously studied residue water content values, which had been dried and stabilized [10,44]. The remaining fraction was attributed to residual carbohydrates from the fruit pulp or seeds present in the residue, consistent with other authors’ reported results [44,45,46,47]. Slight differences can be attributed to variations in drying conditions and the specific characteristics of the plants, such as their origin, variety, and final composition [8]. The SEM images confirmed that fibers had tubular morphology, with a smooth surface and very regular structures (Appendix A).

### 3.2. Physicochemical Characterization of the Laminates

Adhesive viscosity plays a crucial role in determining application performance. It reflects the adhesive capacity of the formulation, making it essential to study this property. The adhesive exhibited a pseudoplastic behavior, meaning that the shear viscosity of adhesives decreased with the increase in the shear rate, with an apparent viscosity at 500 1/s of 30.9 mPa.s that satisfactorily fitted the Ostwald de Waele model [24]. According to Cai et al. [37], shear thinning improves adhesive application and substrate penetration.

With the implementation of thermo-compression processing, it was feasible to obtain laminates from the industrial rosehip oil extraction residue (R) as a substrate and cassava-starch-based adhesive.

Regarding the pressing temperature, a minimum temperature of 120 °C was required for the formation of a thermoplastic starch matrix within the adhesive [24,48]. Figure 2 shows the influence of the processing temperature, with a fixed time of 3 min, on the appearance of the materials obtained by thermo-compression. It was observed that the laminates presented the characteristic color of the residue, orangish with hue values in the range of 66.4–70.7, which intensified as the processing temperature increased. A similar trend was observed when processing times were increased, with significantly higher (*p* ˂ 0.05) color difference values (ΔE) among the samples (Table 2). Such differences can be associated with a higher browning of the fibers that compose the rosehip residue.

The integrity of the materials remained satisfactory regardless of the processing temperature, resulting in a proper mixture of the adhesive with the residue to form a cohesive and uniform structure without observable pores or cracks (Figure 2). In line with these findings, Jiménez [49] stressed that an excessive amount of adhesive not only represents a potential cost escalation but also does not invariably translate into increased material strength. In this sense, the authors elucidate that the incorporation of adhesive ranging between 6% and 10% of the dry weight after processing, as achieved in this study, is required to reach optimal bonding between components.

As shown in Figure 2, all surface images and cross-sections showed that the starch-based adhesive allowed to bind the rosehip fluff residue. These results could be explained by the fact that processing at higher temperatures facilitates the R fibers’ packing promoted by the cross-linking action of the CA [50]. This polyfunctional compound has the capability to form covalent bonds with polymeric chains, resulting in the creation of cross-linked structures. According to Cai et al. [37], the hydrophilic groups present in citric acid and lignocellulosic compounds participate in a cross-linking reaction subsequent to hot-pressing, facilitating the formation of a robust cross-linked network. This cross-linking mechanism following thermal treatment is, therefore, critical for the enhancement of cohesive properties within the material matrix. Material thickness showed significant variability in relation to processing parameters, with reduced values observed at higher temperatures and extended thermo-compression times. Similarly, the moisture content exhibited an inverse relationship with processing time, attributable to greater water evaporation occurring during the extended thermal exposure period (Table 2).

Likewise, ATR-FTIR was used to confirm the interactions between the compounds of the system in the presence of CA. The spectra of the laminates are depicted in Figure 3.

The spectra of the rosehip oil extraction residue (R) showed a broad absorption band in the range of 3750–3000 cm^−1^ attributed to the stretching of O-H bonds, which was sustained through the incorporation of cassava-starch-derived adhesive for laminate formation. However, a lower intensity of this band was observed after the materials were obtained (Figure 3). According to Hassani et al. [51], the rearrangements and the formation of new hydrogen bonds were responsible for the decrease in the band intensities for the hydroxyl groups around 3000 cm^−1^.

The bands observed at 2850 and 2922 cm^−1^ were ascribed to symmetric and asymmetric C–H stretching vibrations in lignin, cellulose, or hemicellulose components [52]. The peak at 1725 cm^−1^ was related to carbonyl groups associated with lignin, and the band at 1642 cm^−1^ was associated with the absorbed and conjugated C–O bonds, while the band at 1595 cm^−1^ was attributable to the aromatic skeletal vibration of lignin. In laminates, the absorption band at 1725 cm^−1^ belonged to C–O stretching characteristic of carbonyls, such as carboxylic acids and esters, supported the chemical linkages between the carboxyl groups of the CA and the free hydroxyl groups of the starch and the lignocellulosic compounds of R. As reported by Chen et al. [53], the slight variation in the peak at 1595 cm^−1^ suggested a possible loss of C=C bonds associated with the aromatic structure, indicating that cross-linking likely occurred between aromatic units in lignin as the temperature increased. This behavior was more pronounced in the laminates produced at 130 °C compared to those obtained at 120 °C (Figure 3a,b).

Since these trays were designed for future applications in food packaging, surface roughness was evaluated. The parameters associated with the roughness of the trays are presented in Table 2. Both the arithmetic mean roughness (Ra) and average roughness (Rz) values obtained were within the range of those reported by Ulker [54]. A significant decrease (*p* < 0.05) was observed in both roughness parameters of the processed material at 120 °C, 3 min (Table 2), potentially attributable to better packing of the rosehip by-product particles during material consolidation.

The surface wettability of materials provides pertinent information to characterize the hydrophobicity of the matrix and its susceptibility to high-humidity environments. In general, the surfaces of the trays were found to be hydrophobic, with contact angles greater than 85°. The materials obtained at a higher thermo-compression temperature presented higher contact angles compared to those obtained at low temperature (Figure 4), leading to the formation of water-resistant matrices. These results could be explained by the cross-linking capacity of CA [24,35] due to the greater availability of carboxyl groups to react with the hydroxyl groups present in both the starch and the lignocellulosic compounds from R. Furthermore, the decline in surface wettability was linked to a reduction in intermolecular hydrogen bonding interactions and an increase in covalent bonds and hydrophobic interactions promoted by the thermal treatment. In addition, R residual lipids and the high content of lignin (Table 1) contributed to the composite materials’ hydrophobicity. In agreement with the results reported by Hassani et al. [51], lignin in unbleached cellulose from biowaste acted as a cementing material among the fibers, leading to more dense and compact structures that hindered water absorption. It is worth noting that an increased contact angle correlated with an enhanced liquid–liquid interaction strength, rendering the material more hydrophobic [55].

Likewise, Figure 5 shows the effect of thermo-compression conditions on the puncture fracture resistance of the materials. The materials thermoformed at 120 °C presented a significant decrease (*p* < 0.05) of the maximum puncture stress resistance with increasing processing time (Figure 5) that could be associated with the observed thickness reduction (Table 2). A similar trend was observed for toughness (Figure 5). Mireles et al. [26] have reported that the presence of fibers in the tray obtention may hinder expansion during thermoforming, creating a discontinuity between the starch chains, thereby compromising the matrix integrity. However, this effect is counteracted by the presence of CA and the cross-linking reactions promoted by this polycarboxylic acid. Notably, trays produced under conditions of 130 °C, 3 min exhibited higher stress values, with a significant increase (*p* < 0.05) in both maximum stress resistance and toughness compared to 120 °C, which could be promoted by the cross-linking action of the CA and the formation of thermoplastic starch. However, no significant effect (*p* > 0.05) was observed under longer pressing times (130 °C, 5 min).

The possible cross-linking of CA with cellulose starts with the dehydration of CA to form a cyclic anhydride, followed by esterification with the hydroxyl groups of the cellulose/hemicellulose or starch. At temperatures higher than 120 °C, CA is partially dehydrated to form citric anhydrides, which are even more reactive toward hydroxyl groups [56,57,58]. These anhydrides readily form ester linkages with polysaccharides, increasing the efficiency of the cross-linking process during thermal treatment. As regards the effect of processing time at 130 °C, it can be observed that thinner materials were obtained at higher times. While material formation was achievable under all conditions tested, a significant improvement in material integrity and both mechanical and water resistance was observed at 130 °C as a result of the cross-linking between the bioadhesive and substrate. Consequently, the selected condition for obtaining multilayer materials was 130 °C, 3 min (L), considering that shorter processing times lead to lower energy requirements.

These laminates obtained were subsequently used as core materials for the design of multilayer systems formed by coating with PLA films assembled by thermo-compression. In this regard, the simplicity and lower environmental impact of processing PLA through the thermo-compression technique compared to solvent processing with chloroform should be highlighted [41]. For this purpose, semi-transparent thermoformed PLA films with a mean thickness of approximately 253 ± 23 µm were obtained in this study.

### 3.3. Characteristics of the Developed Multilayer Materials

The multilaminate trays exhibited uniform coloration and a visually appealing appearance, which indicated that the formulation thermo-compression stage was satisfactory (Figure 1). Thermo-compression proved effective in facilitating interlaminar adhesion, yielding laminates with consistent layer thicknesses. The control laminates (L) presented an average thickness of 761 ± 17 µm. Upon the addition of PLA films on both sides (Figure 1), a markedly higher value of 862 ± 17 µm was observed. Given the compatibility between the substrates, the PLA film demonstrated uniform and efficacious adhesion across the panel surface during the thermo-compression stage. The temperatures achieved in the thermo-compression process induced PLA melting, which facilitated the interaction with the substrates.

Similarly, when the PLA+CP+PLA system was placed in the press, a thickness of 1032 ± 47 µm was obtained. This increase in thickness may stem from the restriction of paste from spreading due to PLA containment. Table 3 illustrates the influence of processing conditions (single or two-stage pressing) on material density as well. Low density is desirable, as it contributes to reduced production and transport costs. According to Pornsuksomboon et al. [59] and Kaisangsri et al. [60], density is one of the most important physical properties when it comes to the practical application of trays. These density values were higher than that reported for conventional polystyrene (PS) trays, 0.03 ± 0.001 g/cm^3^ [10]. However, comparative lifecycle analysis studies have shown that biodegradable materials have lower greenhouse gas emissions and lower consumption of fossil resources compared to conventional PS. As indicated by Ingrao et al. [61] in a lifecycle analysis of meat packing trays, the major environmental impact of these plastic-based products is related to the non-renewable energy sources’ depletion and climate change associated with greenhouse-effect gas emissions derived from the resin production. Moreover, Razza et al. [62] have demonstrated that starch-based materials require 50% less non-renewable energy during production and generate 60% fewer greenhouse gas emissions than conventional expanded PS containers. Additionally, biodegradable polymers offer a sustainable solution for applications like food containers, where recycling is challenging or impractical, thereby helping to reduce solid plastic waste generation.

As expected, the PLA coating of composite biobased materials enhanced moisture resistance, reducing water content, absorption capacity, and solubility (Table 3).

The latter has a clear disadvantage when littering and waste overflows to water streams are considered. In this regard, the PLA+L+PLA trays presented the lowest solubilization capacity in water, being almost four times lower than control laminates (L). These results can be attributed to the fact that PLA waterproofs the material by acting as an external barrier that prevents water from penetrating the more hydrophobic core. The PLA layers act as an outer layer to prevent water from ingressing and provide additional strength to the material. Similarly, Wang et al. [63] emphasized that the addition of PLA significantly improved the physical and mechanical properties of corn starch–polylactic-acid-based films, counteracting the shortcomings of each component of the system.

Complementary studies by SEM showed that the cross-section and the surface of the PLA film were smooth, without holes and cracks (Figure 6). Regardless of the system obtained by SEM, the compatibility at the microstructural level of the PLA films with the paste or with the laminate was evident. During the thermo-compression of the PLA+CP+PLA matrix, the PLA film was intercalated and incorporated into the fibrillar structure characteristic of the residue (Appendix A), as we observed in the cutting section of the material (Figure 6b). Likewise, there was an adequate interaction between substrates because the matrix was embedded in the film. The CP components reacted with each other, but the formation of the network had interrupted sites with the PLA in the molten state.

On the other hand, the thermoformed materials subjected to two heat pressure treatments prevented the PLA film from penetrating as much into the material (Figure 6c). Nonetheless, it made an adequate coating, with no recorded pores or cracks, which corroborated that the interactions were at the interface level of the layer.

Regarding the surface properties, the laminated material (L) presented higher medium roughness than those coated with PLA, attributed to the low roughness of the PLA films (Ra = 3 ± 0.4 μm; Rz = 9 ± 1 μm). These results agree with the SEM surface observations (Figure 6 on the right).

Moreover, both the PLA film and PLA+L+PLA trays showed higher average values of 92 ± 3°, with no significant differences with the composite L, all corresponding to hydrophobic surfaces. Nevertheless, an increase in the wettability of the materials was observed in the PLA+CP+PLA system, presenting an average contact angle value of 82 ± 3°. This phenomenon may be attributed to specific processing parameters that potentially altered the core temperature profile of the composite matrix during thermoforming operations, thereby potentially impeding citric acid cross-linkage mechanisms and, consequently, yielding a material with enhanced hydrophilicity, as was evidenced in laminated structures subjected to reduced thermal conditions (Figure 4). Besides, the in situ cross-linking of the bioadhesive in the CP within the PLA films led to a greater intercalation of PLA within the composite matrix observed by SEM (Figure 6b).

Nonetheless, these could lead to some exposure of the cellulosic and more hydrophilic filler fibers, leading to an increase in wettability and water resistance (Table 3). The superficial exposure of these fibers was identified with the increase in the FTIR signal at around 2850 and 2922 cm^−1^, ascribed to symmetric and asymmetric C–H stretching vibrations in lignin, cellulose, or hemicellulose components (Figure 7).

As is known, PLA is a hydrophobic polymer because of the presence of −CH side groups. The peaks at about 2997 and 2946 cm^−1^ corresponded to the asymmetric and symmetric stretching vibrations of the −CH groups in the side chains (methyl group in PLA structure), respectively. A prominent peak in the PLA spectrum was observed at 1745 cm^−1^, which represented the stretching of the amorphous phase of carbonyls. The absorption band assigned to the C–H deformation vibration was detected at 1456 cm^−1^ and 1384 cm^−1^. Additionally, the bands located at 1180 and 1080 cm^−1^ belonged to the asymmetric and symmetric stretching vibrations of C−O−C, which were detected. The symmetric and asymmetric −CH_3_, -CH, and C–C stretching vibrations were observed at 1047 cm^−1^, 955 cm^−1^, and 870 cm^−1^, respectively (Figure 7).

In multilayer materials, regardless of the production route, the appearance of the characteristic peaks of PLA indicated the presence of this polymer, acting as an outer layer. These findings correlated with surface properties of the composite materials, similar to neat PLA film properties.

Mechanical profiles obtained for trays formulated by the two proposed routes (Figure 1) are shown in Figure 8. It was observed that the incorporation of PLA in the laminates led to a notable increase in maximum puncture stress values, resulting in harder and more resilient materials. The PLA film formed a thin, homogeneous layer along the laminate, which indicated good compatibility in the matrix. These observations were verified by SEM micrographs (Figure 6).

Ferreira et al. [8] reported that the compaction process of the material formation induced a reduction in porous interstices between residue particle aggregates, accompanied by matrix rearrangement. This phenomenon was pressure dependent. Consequently, the orientation and rearrangement of the residue particles facilitated the formation of a cohesive mass within the tray-like material matrix due to the hot-pressing treatment. Furthermore, cross-linking reactions between the starch components partially hydrolyzed by CA [24,35] and the rosehip by-product components could have been promoted. All the above may have contributed to an enhanced mechanical resistance of the multilaminate materials, making this effect stronger for the two-step process (Figure 8a,b). All in all, the double-pressing treatment used to obtain the PLA-L-PLA material would explain the better mechanical resistance due to the cross-linking by the action of citric acid.

It should be noted that puncture testing allows a comparative measurement of the mechanical behavior of the developed materials. Further mechanical testing assays, such as the tensile, impact, and bending resistance of the trays, shall be conducted after pilot-scale processing parameters are optimized in standard production lines.

## 4. Conclusions

Sustainable trays with adequate mechanical properties were successfully formulated using the industrial waste derived from rosehip fruit essential oil extraction and bioadhesives via thermo-compression, making a novel advancement hitherto unexplored in the literature. The development of multilayer materials compensates for the shortcomings of the formulated laminate composite biomaterials with adhesive and residue, thus improving their suitability for food packaging.

The cross-linking reaction between starch, citric acid, and the residue particles induced the formation of a complex network by thermo-compression, thereby improving the adhesive performance for starch-based materials. Cross-linking was evident, obtaining rather hydrophobic matrices at treatments conducted at 130 °C.

Furthermore, a strategic approach to improving the mechanical properties of the composite materials was proposed, involving the fabrication of multilaminate systems’ production through thermo-pressing with PLA films. The incorporation of a PLA film allowed the development of mechanically suitable, moisture- and puncture-resistant trays through a simple and solvent-free thermo-compression process. While the two-step lamination offered better mechanical and water resistance, it required more time and energy. In contrast, single-step thermo-compression reduced processing costs and time demand, making it more sustainable, though with slightly lower performance. Still, this method outperformed traditional laminated materials. The choice between multilayer formats should depend on the packaging needs of the product. These trays offer a promising alternative to polystyrene, with potential applications in the packaging of dry and semi-moist food products.

When choosing packaging materials, there is a fundamental tradeoff between immediate economic factors and long-term environmental considerations. Expanded polystyrene offers lower upfront costs per unit, making it initially attractive for businesses focused on minimizing packaging expenses. However, the additional costs can be offset in the long run by environmental benefits. In contrast, biopolymers such as PLA and starch can degrade under composting conditions.

While the present investigation has established fundamental characterization parameters, subsequent research endeavors will address additional requisite assessments for food-contact applications, including comprehensive migration analyses with appropriate food simulants and thermal stability evaluations pertinent to potential sterilization protocols. These critical investigations represent essential extensions of the current work to fully validate the developed trays for their intended commercial applications.

Overall, this study demonstrated the feasibility of using agri-food residues to develop sustainable packaging solutions, contributing to the transition toward a circular economy in the packaging industry.

## Figures and Tables

**Figure 1 foods-14-01843-f001:**
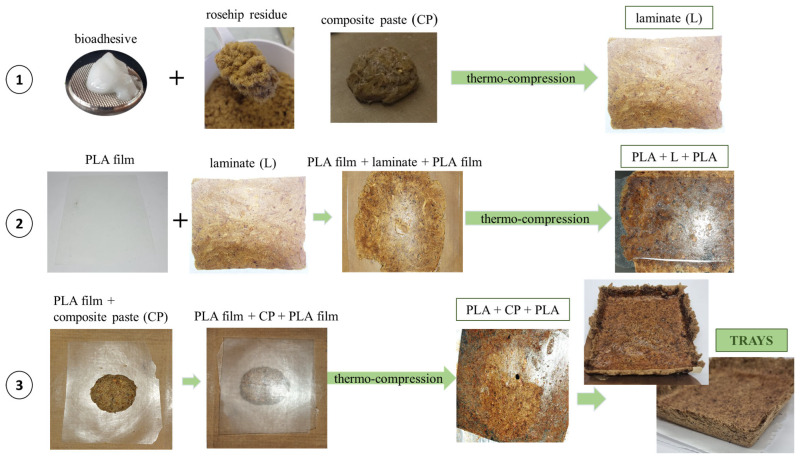
Schematic representation of obtaining the trays: (1) rosehip residue sustainable laminated material (L) processing, (2) multilayered material preparation from laminate and PLA films (PLA+L+PLA), and (3) in situ thermoforming of the composite material within PLA film layers (PLA+CP+PLA).

**Figure 2 foods-14-01843-f002:**
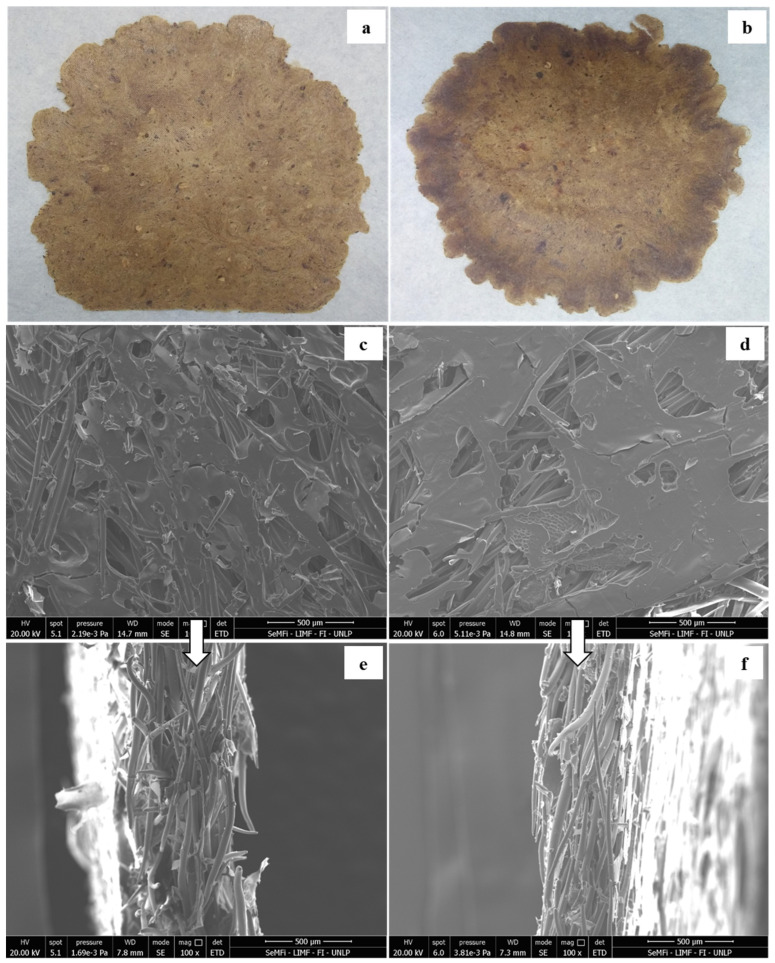
Materials based on bioadhesive formulations with citric acid and by-products of rosehip essential oil obtained by thermo-compression for 3 min at 120 °C (left) and 130 °C (right). Top to bottom: laminate photograph (**a**,**b**), SEM micrographs (100×) of materials’ surfaces (**c**,**d**), and cryo-fractured cross-sections (**e**,**f**).

**Figure 3 foods-14-01843-f003:**
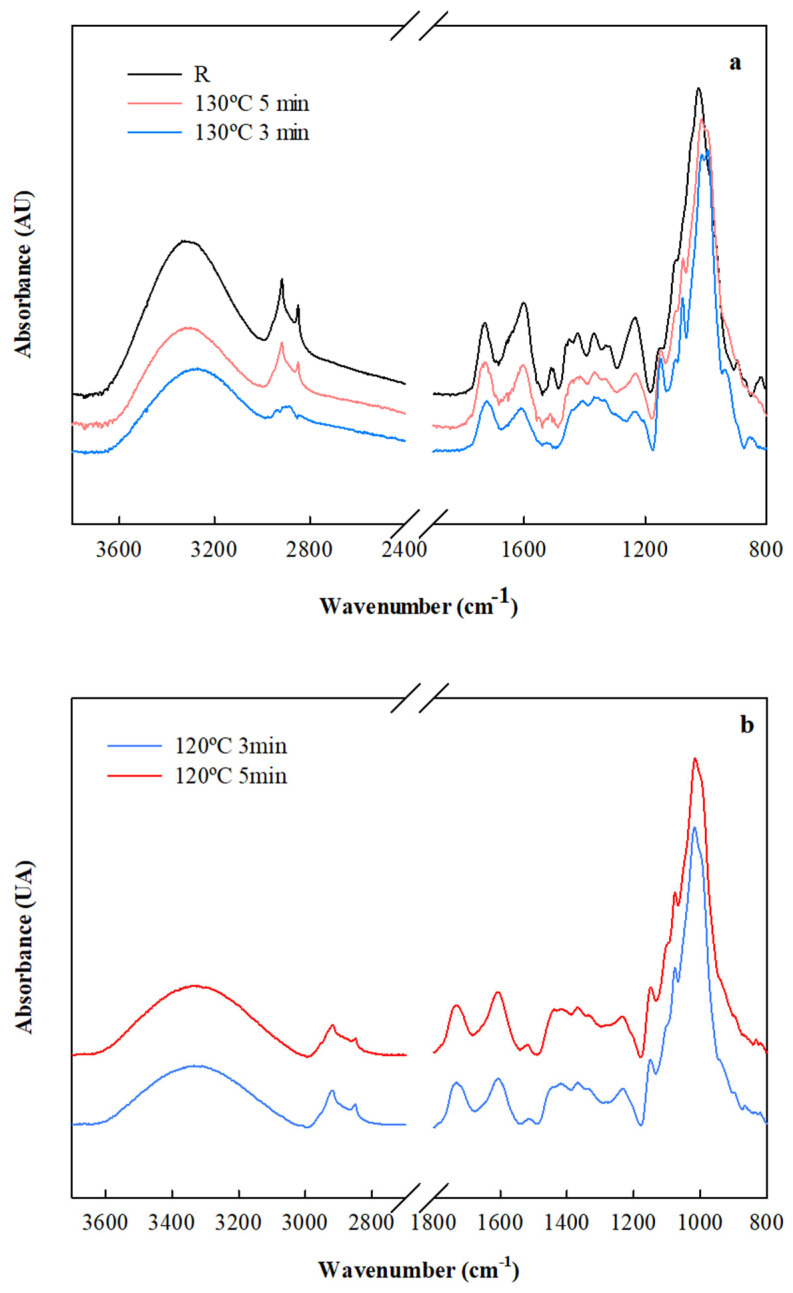
ATR-FTIR spectra of the R by-product and laminated materials obtained through thermo-pressing at (**a**) 130 °C, (**b**) 120 °C, and different processing times (3 and 5 min).

**Figure 4 foods-14-01843-f004:**
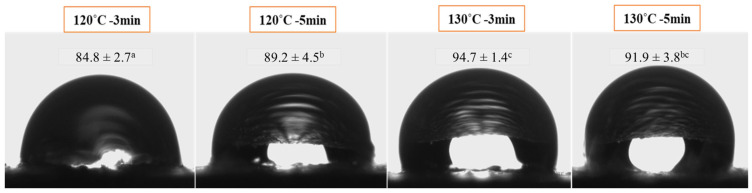
Photographs of water drops deposited on trays based on R waste and bioadhesives. Reported contact angles correspond to the mean ± standard deviation. Different letters indicate significant differences (*p* ˂ 0.05).

**Figure 5 foods-14-01843-f005:**
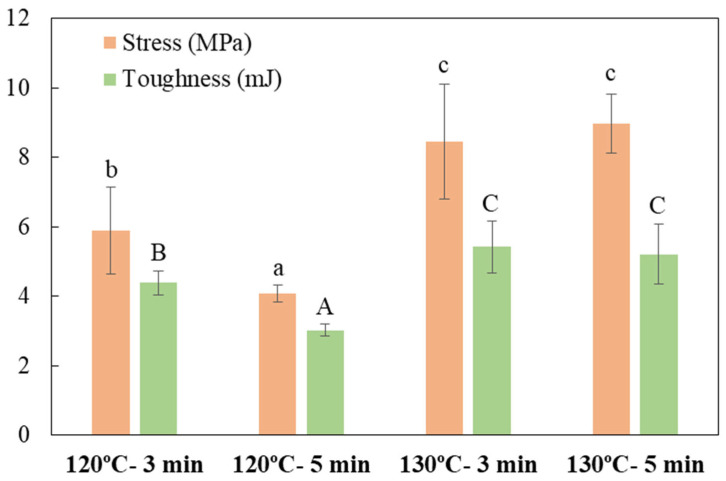
Influence of thermo-compression conditions (temperature/time) in the formation of laminates on maximum puncture stress (MPa) and toughness (mJ). Different lowercase letters indicate significant differences in stress values, while different uppercase letters indicate significant differences between toughness parameters (*p* < 0.05).

**Figure 6 foods-14-01843-f006:**
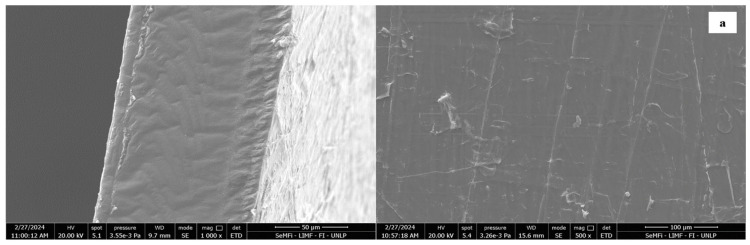
SEM micrographs of the cross-section (left) and surface (right) of materials obtained by thermo-compression: (**a**) PLA film, (**b**) PLA+CP+PLA, and (**c**) PLA+L+PLA. The magnification used is indicated in the micrographs.

**Figure 7 foods-14-01843-f007:**
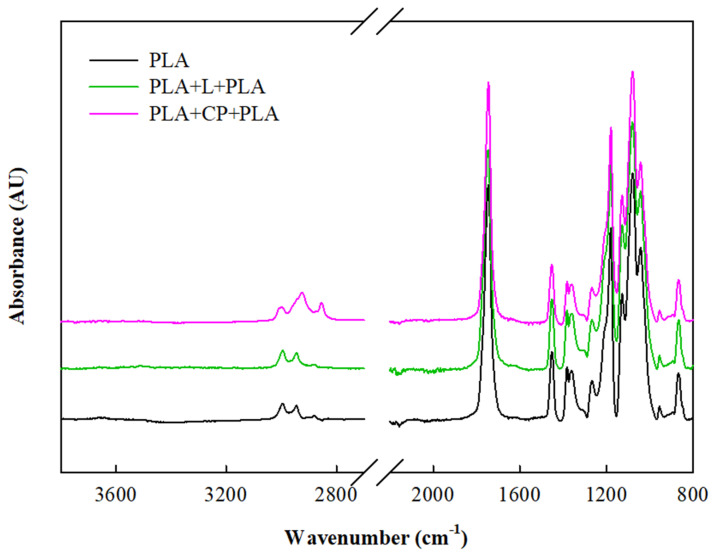
ATR-FTIR spectra of single PLA film and multilayered materials through single-step (PLA+CP+PLA) and two-step (PLA+L+PLA) processes.

**Figure 8 foods-14-01843-f008:**
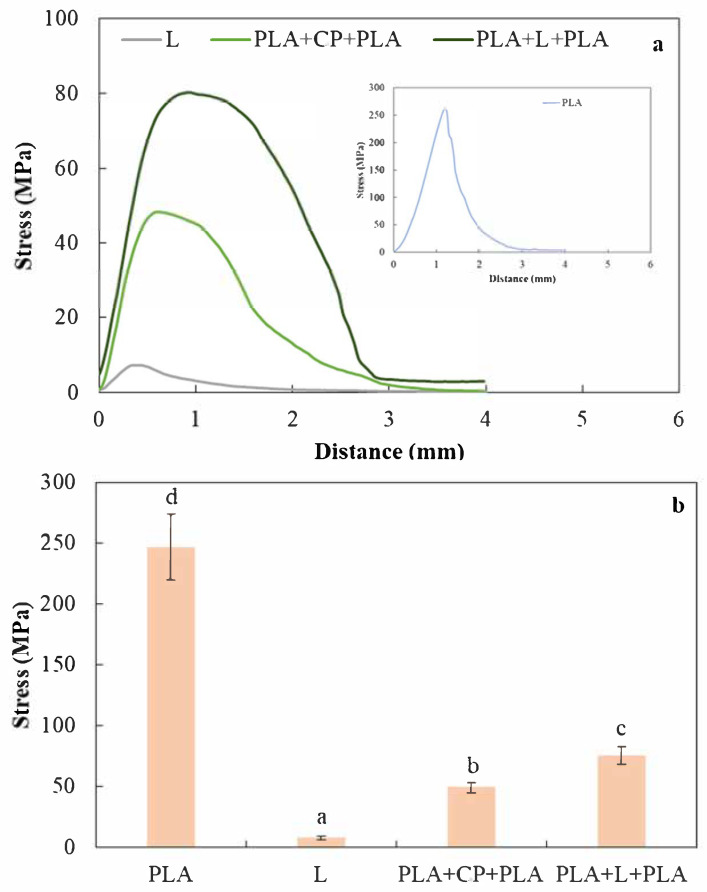
(**a**) Mechanical profiles obtained in a puncture test and (**b**) maximum stress values recorded for single thermoformed PLA films (PLA), composite biobased laminated materials (L), and multilayered materials obtained through single-step (PLA+CP+PLA) or double-step (PLA+L+PLA) processes.

**Table 1 foods-14-01843-t001:** Chemical composition of rosehip fluff residue (% wet basis).

Moisture	Total Fiber			Ashes	Proteins	Lipid
	Cellulose	Hemicellulose	Lignin			
9.91 ± 1.3	23.84 ± 1.3	11.63 ± 2.3	28.21 ± 0.6	2.63 ± 0.1	1.69 ± 0.2	0.79 ± 0.1

The average values ± standard deviation are shown.

**Table 2 foods-14-01843-t002:** Physicochemical parameters of the trays obtained by thermo-compression.

Sample	ΔE	Thickness (µm)	Moisture (%)	Roughness Parameters (μm)	
				Ra	Rz
120 °C, 3 min	48.2 ± 1.9 ^a^	819 ± 43 ^c^	3.48 ± 0.08 ^a^	10.9 ± 0.9 ^a^	30.9 ± 2.6 ^a^
120 °C, 5 min	52.8 ± 1.9 ^c^	723 ± 40 ^a^	4.23 ± 0.01 ^b^	15.9 ± 1.9 ^b^	47.9 ± 5.0 ^b^
130 °C, 3 min	52.5 ± 0.9 ^c^	761 ± 17 ^b^	3.54 ± 0.05 ^a^	22.5 ± 2.3 ^c^	63.6 ± 6.5 ^c^
130 °C, 5 min	50.4 ± 0.7 ^b^	707 ± 30 ^a^	4.68 ± 0.44 ^b^	20.6 ± 2.5 ^c^	60.1 ± 4.5 ^c^

Mean ± standard deviations are presented. Different letters in the same column indicate significant differences (*p* ˂ 0.05).

**Table 3 foods-14-01843-t003:** Physicochemical properties, water affinity, and resistance of trays based on cassava starch adhesive and rosehip reinforced with PLA film.

Properties/Samples		L	PLA+CP+PLA	PLA+L+PLA
ΔE		52.5 ± 0.9 ^b^	49.3 ± 0.7 ^a^	55.4 ± 0.8 ^c^
Ra (μm)		22 ± 3 ^a^	17 ± 6 ^a^	16 ± 6 ^a^
Rz (μm)		64 ± 6 ^a^	48 ± 17 ^a^	45 ± 17 ^a^
Density (g/cm^3^)		0.48 ± 0.005 ^a^	0.93 ± 0.02 ^b^	1.05 ± 0.003 ^c^
WAC (%)	1 min	39.7 ± 1.3 ^c^	5.8 ± 0.08 ^b^	3.5 ± 0.1 ^a^
	60 min	104 ± 13 ^c^	19 ± 1 ^a^	22 ± 1 ^b^
Moisture (%)		6.2 ± 0.1 ^c^	3.3 ± 0.1 ^b^	2.5 ± 0.1 ^a^
Solubility (%)		20.6 ± 1 ^c^	11.9 ± 0.9 ^b^	5.9 ± 0.4 ^a^
Contact angle (°)		94 ± 1 ^b^	82 ± 3 ^a^	92 ± 3 ^b^

Mean ± standard deviations are presented. Different letters in the same file indicate significant differences (*p* ˂ 0.05). Composite biobased laminated materials (L) and multilayered materials obtained through single-step (PLA+CP+PLA) or double-step (PLA+L+PLA) processes.

## Data Availability

The original contributions presented in this study are included in the article/Appendix A. Further inquiries can be directed to the corresponding author.

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
