# Peer review of "Ecological Packaging: Reuse and Recycling of Rosehip Waste to Obtain Biobased Multilayer Starch-Based Material and PLA for Food Trays"

_foods, 2025, doi:10.3390/foods14111843_

Round 1
Reviewer 1 Report
Comments and Suggestions for Authors
Comments from Reviewer
Manuscript ID: Foods-3616247
Title: Ecological packaging: reuse and recycling of rosehip waste to obtain starch-based trays
On the other hand, I have found some points of this manuscript that the authors must take into account these are indicated in yellow in the pdf document. Moreover, I have also performed the review and my comments on each section are described as follows:
Abstract.
- In my opinion, the abstract content describes in general terms the context of the research performed by the authors.
- Some words have some grammatical mistakes, which were highlighted in the PDF manuscript. The authors use capital letters in words that are unnecessary to write in these letters.
- It is recommended that sentences be direct and concise.
- Is the term “bioadhesive” correct? As for example, “compatibilizer” or other? Authors are advised to use the terms correctly, to avoid mistakes.
- The last sentence of this section is significant because it presents the perspective of the application or the problem to be solved with this research or material.
Introduction
- To start, this section has several grammar mistakes, many of which are related to misspelling words, missing articles, or incorrect prepositions. Moreover, this section can be observed wordy and unclear sentences, passive voice misuse, and hard-to-read text. Several sentences have punctuation mistakes.
- Generally, when quantification of objects or actions is done, a comma is placed before the last “one, two, three, and four”
- The authors must avoid using colloquial words; it is a scientific manuscript. For example, “ …. the conversion of hydroxyl (OH) polymers …” for better clarity of your sentences.
- Several sentences in the manuscript contain very long sentences (more than three lines); some of these sentences lack punctuation marks, so the ideas intended to be expressed in the sentence are not separated.
- The introduction describes the main topics to be developed in this manuscript. However, several points need to be clarified. For example, what is the purpose of developing a multi-layer semi-rigid assembly? What is the benefit to the physicochemical and performance properties of the material? These questions are not stated in this section.
- The last paragraph of this section is very important because the reader can understand the aims of this research. The authors described the aims of this research, which are not clearly stated in the manuscript. It should be written in the last paragraph of this section.
Materials and Methods and Results
- Several mistakes have been highlighted in yellow in these sections. In general, the most common mistakes are change of preposition, verb, incorrect use of an article, article missing, name misspelling, etc. I just only highlighted the mistakes observed in sections 2 and 3, the others sections are only revised in their content. The manuscript must be revised by a native English speaker.
- Use homogeneously the notation of the units “w/v” and avoid “mm min-1”, which are very different ways of placing the units. Authors should take care of the homogeneity of the units in the manuscript.
- Moreover, in a scientific text, the use abbreviation according to SI units must be used. For instance, minutes is min, hours is h, etc. These were highlighted in the manuscript.
- The term “L” (page 4, line 137) is confused with the term “L” (page 5, line 163), it should be modified.
- The element count should correctly be “one, two, and three”.
- What is the residue content in the composites? It is important to point out the biomass and bioplastic content in each of the two types of composites produced. This information is essential for the comparative analysis of the biocomposites produce.
- Which are the analyses to evaluate the quality of the product developed?
- What kind of requirements must follow the tray developed? In my opinion, the test to characterize the trays is not complete. For instance, the migration test is important if the tray will be in contact with food.
- Thermal properties are very important to be determined in this kind of materials, por possible sterilization process in products to be in contact with food.
Results and Discussion.
- Topics and subtopics are NOT separated in the manuscript. Authors should place them in the order in which they are presented in the Materials and Methods section. Authors should adhere to the Journal Author Guidelines before submitting a manuscript to a journal. Authors should check how figures are cited. They are cited as if they were supplementary material. Check the Journal's Author's Guide.
- Table 1 does not have the units of the parameters reported. In what units are you reporting cellulose, lignin, proteins, lipids, etc.? They do not find the results for soluble fiber and insoluble fiber of this agroindustrial residue.
- In summary, the FTIR spectra are a repeat of the PLA spectra in the formulations made. What is the effect of analyzing the FTIR spectra, when it is well known that the infrared laser beam only penetrates a small part of the material and by logic it will be the PLA matrix that will appear in the spectrum.
- On the other hand, there are several errors in the wavenumber units in the manuscript that should be corrected.
- The morphology of the SEM micrographs corroborates the comment of the PLA matrix, which coats the material.
- In Table 3, how can the authors argue the difference between the contact angle values of PLA+CP+PLA and PLA+L+PLA biocomposites?. Similar question is formulated to the parameters Solubility and VAC at 1 min.
- The authors should review the discussion they present on various topics. It is shallow and lacking in rigorous scientific argumentation. It is important to point out that it is a scientific publication, not a technical report.
- In my opinion, it seems to me that the work lacked experiments to argue or support the application of the biocomposite they are producing. Is the biocomposite susceptible to puncture damage? Does the food product have those characteristics that cause a puncture fracture in the material?
- The word spelling in several sentences must be revised. Avoid colloquial expressions in your manuscript
Conclusion
- I strongly suggest that the authors revise and re-write some statements. Several sentences in this section are very general (observations) and do not provide a conclusion. My suggestion is: The conclusion section provides an objective analysis of the literature and their main findings or results the evidence from the literature supports the statements.
- The authors wrote the following: The cross-linking reaction between starch, polycarboxylic acid and the residue particles induced the formation of a complex network by thermo-compression, thereby improving the adhesive performance for starch-based materials. However, in their abstract, introduction, and section 2.3. they do not mention the use of polycarboxylic acid. This and other inconsistencies can be noted in the manuscript.
- Some writing issues in this manuscript include word choice, punctuation in compound/complex sentences, wordy sentences, unclear sentences, incorrect prepositions, word misspellings, missing articles, etc.
The authors must revise the references; there are several mistakes, and the information is not complete (volume, pages, etc.). Some references have DOIs, and others do not. Authors should homogenize their references following the authors' guide guidelines. As already mentioned several times in this review.
My final comment is that this manuscript cannot be accepted for publication. It is rejected, and the authors must improve several sections, discuss some sections, and re-write the parts listed in this manuscript. A native speaker must revise English; I found several mistakes in the sentences written by the authors.
Sincerely,
The reviewer

Comments on the Quality of English Language
A native speaker must revise English; I found several mistakes in the sentences written by the authors.
Author Response
We have done the revision of our manuscript following the Reviewers` suggestions. We would like to thank the Reviewers for their appropriate and useful comments and suggestions that help us to improve the manuscript quality. Accordingly, the suggested changes were introduced in the revised version in blue, in order to highlight them.
Comment 1: Abstract section:
- In my opinion, the abstract content describes in general terms the context of the research performed by the authors.
- Some words have some grammatical mistakes, which were highlighted in the PDF manuscript. The authors use capital letters in words that are unnecessary to write in these letters.
- It is recommended that sentences be direct and concise.
- Is the term “bioadhesive” correct? As for example, “compatibilizer” or other? Authors are advised to use the terms correctly, to avoid mistakes.
- The last sentence of this section is significant because it presents the perspective of the application or the problem to be solved with this research or material.
Answer: We appreciate the reviewers comments. The abstract has been improved following your suggestions.
Comment 2: Introduction section
- To start, this section has several grammar mistakes, many of which are related to misspelling words, missing articles, or incorrect prepositions. Moreover, this section can be observed wordy and unclear sentences, passive voice misuse, and hard-to-read text. Several sentences have punctuation mistakes.
- Generally, when quantification of objects or actions is done, a comma is placed before the last “one, two, three, and four”
- The authors must avoid using colloquial words; it is a scientific manuscript. For example, “ …. the conversion of hydroxyl (OH) polymers …” for better clarity of your sentences.
- Several sentences in the manuscript contain very long sentences (more than three lines); some of these sentences lack punctuation marks, so the ideas intended to be expressed in the sentence are not separated.
- The introduction describes the main topics to be developed in this manuscript. However, several points need to be clarified. For example, what is the purpose of developing a multi-layer semi-rigid assembly? What is the benefit to the physicochemical and performance properties of the material? These questions are not stated in this section.
- The last paragraph of this section is very important because the reader can understand the aims of this research. The authors described the aims of this research, which are not clearly stated in the manuscript. It should be written in the last paragraph of this section.
Answer: Thank you very much for the observations. English grammar has been thoroughly checked.
Comment 3: Materials & Methods section
- Several mistakes have been highlighted in yellow in these sections. In general, the most common mistakes are change of preposition, verb, incorrect use of an article, article missing, name misspelling, etc. I just only highlighted the mistakes observed in sections 2 and 3, the others sections are only revised in their content. The manuscript must be revised by a native English speaker.
Answer: Thank you very much for the observations. English grammar has been thoroughly checked.
- Use homogeneously the notation of the units “w/v” and avoid “mm min-1”, which are very different ways of placing the units. Authors should take care of the homogeneity of the units in the manuscript.
- Moreover, in a scientific text, the use abbreviation according to SI units must be used. For instance, minutes is min, hours is h, etc. These were highlighted in the manuscript.
- The term “L” (page 4, line 137) is confused with the term “L” (page 5, line 163), it should be modified.
- The element count should correctly be “one, two, and three”.
Answer: Thanks for the observations, all the Reviewer's suggestions were considered in the revised version.
- What is the residue content in the composites? It is important to point out the biomass and bioplastic content in each of the two types of composites produced. This information is essential for the comparative analysis of the biocomposites produce.
Answer: Thanks for your observation, the residue content in the composite was added as follows: “Based on previous studies, the R by-product:adhesive ratio was set at 1:1 on a wet basis, resulting in a final content of 90% of filler”
- Which are the analyses to evaluate the quality of the product developed?
- What kind of requirements must follow the tray developed? In my opinion, the test to characterize the trays is not complete. For instance, the migration test is important if the tray will be in contact with food.
- Thermal properties are very important to be determined in this kind of materials, por possible sterilization process in products to be in contact with food.
Answer: Dear Reviewer we agree with your concerns on the subject, however this work was focused on the material compounding and processing to achieve proper mechanical and moisture resistance. This evaluation would be conducted in further studies with food simulants. This will be highlighted in the conclusions
Comment 4: Results and Discussion section
- Topics and subtopics are NOT separated in the manuscript. Authors should place them in the order in which they are presented in the Materials and Methods section. Authors should adhere to the Journal Author Guidelines before submitting a manuscript to a journal. Authors should check how figures are cited. They are cited as if they were supplementary material. Check the Journal's Author's Guide.
Answer: This was corrected. Please note that the supplementary image was added by the system within the manuscript but is expected to be published separately.
- Table 1 does not have the units of the parameters reported. In what units are you reporting cellulose, lignin, proteins, lipids, etc.? They do not find the results for soluble fiber and insoluble fiber of this agroindustrial residue.
Answer: Dear Reviewer, please note that the units are reported in the Tables title. Regarding ADF and NDF results, these were used to estimate cellulose and hemicellulose content as established through standards.
Hemicellulose (%) = NDF (%) - ADF (%)
Cellulose (%) = ADF (%) - Lignin (%)
- In summary, the FTIR spectra are a repeat of the PLA spectra in the formulations made. What is the effect of analyzing the FTIR spectra, when it is well known that the infrared laser beam only penetrates a small part of the material and by logic it will be the PLA matrix that will appear in the spectrum.
- On the other hand, there are several errors in the wavenumber units in the manuscript that should be corrected.
- The morphology of the SEM micrographs corroborates the comment of the PLA matrix, which coats the material.
- In Table 3, how can the authors argue the difference between the contact angle values of PLA+CP+PLA and PLA+L+PLA biocomposites?. Similar question is formulated to the parameters Solubility and VAC at 1 min.
- The authors should review the discussion they present on various topics. It is shallow and lacking in rigorous scientific argumentation. It is important to point out that it is a scientific publication, not a technical report.
- In my opinion, it seems to me that the work lacked experiments to argue or support the application of the biocomposite they are producing. Is the biocomposite susceptible to puncture damage? Does the food product have those characteristics that cause a puncture fracture in the material?
- The word spelling in several sentences must be revised. Avoid colloquial expressions in your manuscript
Answer: We thank the Reviewer for his/her useful comments that help us to improve the manuscript. All the suggestions were carefully considered in the revised manuscript.
Comment 5: Conclusion section
I strongly suggest that the authors revise and re-write some statements. Several sentences in this section are very general (observations) and do not provide a conclusion. My suggestion is: The conclusion section provides an objective analysis of the literature and their main findings or results the evidence from the literature supports the statements.
- The authors wrote the following: The cross-linking reaction between starch, polycarboxylic acid and the residue particles induced the formation of a complex network by thermo-compression, thereby improving the adhesive performance for starch-based materials. However, in their abstract, introduction, and section 2.3. they do not mention the use of polycarboxylic acid. This and other inconsistencies can be noted in the manuscript.
Answer: Even though the information was included in Section 2.3 as a reference to a previous work, we agree with the reviewers that this needed to be clearly explained in this section. We incorporated this information as requested
- Some writing issues in this manuscript include word choice, punctuation in compound/complex sentences, wordy sentences, unclear sentences, incorrect prepositions, word misspellings, missing articles, etc.
Answer: The conclusions were revised and completed according to the reviewers comments. Thank you very much for your thorough and careful critics.
Comment 6: The authors must revise the references; there are several mistakes, and the information is not complete (volume, pages, etc.). Some references have DOIs, and others do not. Authors should homogenize their references following the authors' guide guidelines. As already mentioned several times in this review.
Answer: Thanks for the observation. The refernces were carefully revised following the authors' guide guidelines.

Reviewer 2 Report
Comments and Suggestions for Authors
The study presents the development of polylactic acid (PLA)/starch-based trays incorporating rosehip waste. The use of agricultural waste introduces a degree of novelty and sustainability to the work. However, there are several concerns regarding the experimental design, data interpretation.
- Title: While polylactic acid is a major component in the tray, it is not mentioned in the title.
- Abstract: The abstract is currently too general. Please include key results and quantitative data to support the findings.
- Experimental design: The rationale for including rosehip waste in the tray formulation should be explained more clearly. Is it for functional, nutritional, or structural reasons? What are the potential applications of the starch-based trays? Have any tests been conducted on their use with real food products? Including a photo of the actual tray would be helpful, as Figure 1 does not clearly show the shape or structure. The laminate layer also appears uneven—have the authors considered methods to improve its uniformity?
- Line 127: Please provide a proper reference for the AOAC (1990) standard methods used.
- Figure 1: Suggest revising the label “film PLA” to “PLA film” for consistency and clarity.
- Formatting Issues: There are a few formatting errors throughout the manuscript that need correction. For example:
- Line 159: “2.5. . Characterization of the material” (extra period)
- Line 195: “2.5.2. Microstructural analysis” (inconsistent formatting)
- Line 215–217: The equations used to calculate mechanical properties should be provided in the methodology.
- Figure 5: Please explain the meaning of the different letters (capital and lowercase) used in the figure to indicate statistical differences.
- Statistical Analysis: The assignment of letters (e.g., “a”) to denote statistical significance needs to be clarified and consistently applied across all tables and figures. Typically, “a” indicates the highest value, but this must be clearly stated and follow the logic of the statistical test used.
Author Response
We have done the revision of our manuscript following the Reviewers` suggestions. We would like to thank the Reviewers for their appropriate and useful comments and suggestions that help us to improve the manuscript quality. Accordingly, the suggested changes were introduced in the revised version in blue, in order to highlight them.
Comment 1: Title: While polylactic acid is a major component in the tray, it is not mentioned in the title.
Answer: Following the Reviewer’s suggestion the title was modified as follows: Ecological packaging: reuse and recycling of rosehip waste to obtain biobased multilayer starch-based material and PLA for food trays.
Comment 2: The abstract is currently too general. Please include key results and quantitative data to support the findings.
Answer: We appreciate the Reviewer’s comments. The abstract has been improved following your suggestions.
Comment 3: Experimental design: The rationale for including rosehip waste in the tray formulation should be explained more clearly. Is it for functional, nutritional, or structural reasons? What are the potential applications of the starch-based trays? Have any tests been conducted on their use with real food products? Including a photo of the actual tray would be helpful, as Figure 1 does not clearly show the shape or structure. The laminate layer also appears uneven—have the authors considered methods to improve its uniformity.
Answer: The inclusion of rosehip waste in the tray formulation is primarily for structural purposes. It acts as a biomass-derived filler that provides mechanical support and contributes to forming the packaging material. In addition, modified starch is incorporated as a natural adhesive, binding the particles together and enabling the molding and integrity of the final tray. This combination of agro-industrial residues and biopolymers allows for the development of sustainable, biodegradable packaging materials, aligning with circular economy principles and reducing reliance on synthetic polymers.
Although the tray has a roughness typical of waste use, this is significantly improved by the thermocompression process and the use of PLA film.
The starch-based trays, coated with a polylactic acid (PLA) film that provides moisture barrier properties, have potential applications in the packaging of dry or semi-moist food products.
The combination of biodegradability, structural integrity, and moisture resistance makes these trays suitable for use in sustainable food packaging systems, particularly in contexts where single-use plastics are being phased out. Additionally, a photograph of the obtained trays was included in Figure 1.
Comment 4: Line 127: Please provide a proper reference for the AOAC (1990) standard methods used.
Answer: Thanks for the observation, the reference was added: AOAC (1990). In K. Helrich (Ed.). Official methods of analysis of the association of official analytical chemists(15th ed). Arlington, Va: Association of Official Analytical Chemists.
Comment 5: Figure 1: Suggest revising the label “film PLA” to “PLA film” for consistency and clarity.
Answer: The suggestion has been taken into account, and Figure 1 has been revised accordingly.
Comment 6: Formatting Issues: There are a few formatting errors throughout the manuscript that need correction. For example:
- Line 159: “2.5. . Characterization of the material” (extra period)
Answer: We apologize for the oversight in the original submission. The corresponding sections have been revised, and the necessary modifications have been made in the updated version of the manuscript.
According to the reviewer’s suggestions, the session has been included in the methodology section of the revised manuscript.
- Line 195: “2.5.2. Microstructural analysis” (inconsistent formatting)
Answer: We apologize for the oversight in the original submission. The corresponding sections have been revised, and the necessary modifications have been made in the updated version of the manuscript.
- Line 215–217: The equations used to calculate mechanical properties should be provided in the methodology.
Answer: Answer: The mechanical properties were evaluated following standardized procedures in ASTM D882-00, which were already detailed in the cited previous works. For this reason, and given that other reviewers suggested shortening this section, we prefer to retain the citation.
Comment 7: Figure 5: Please explain the meaning of the different letters (capital and lowercase) used in the figure to indicate statistical differences.
Answer: The suggestion has been accepted, and Figure 5 has been modified. Different lowercase letters indicate significant differences in stress values, while different uppercase letters indicate significant differences between toughness parameters (p < 0.05).
Comment 8: Statistical Analysis: The assignment of letters (e.g., “a”) to denote statistical significance needs to be clarified and consistently applied across all tables and figures. Typically, “a” indicates the highest value, but this must be clearly stated and follow the logic of the statistical test used.
Answer: It's important to note that InfoStat does not automatically assign the letter "a" to the highest mean. The order of letter assignment depends on the sequence of means in the output. To ensure clarity and consistency, we have ordered the means from highest to lowest before applying the multiple comparison test. This approach ensures that the letter "a" consistently corresponds to the highest mean, followed by "b," "c," and so on, reflecting descending order.​
This procedure is consistent with the guidelines provided in the InfoStat manual (Balzarini et al., 2008)
Balzarini, M. G. y Di Rienzo, J. A. 2016. InfoGen. FCA. Universidad Nacional de Córdoba, Argentina. http://www.info-Gen.com.ar.
Balzarini, M. G.; González, L.; Tablada, M.; Casanoves, F.; Di Rienzo, J. A. y Robledo, C. W. 2008. Manual del usuario de InfoStat. Editorial Brujas. Córdoba, Argentina. 348 p.

Reviewer 3 Report
Comments and Suggestions for Authors
Dear Authors,
The Introduction is very detailed, skillfully organized, and well written. It contains only relevant data about the study's topic. The discussion of the obtained results, which are well presented, is excellent and profound. There are comparisons with other studies, as well as a deep explanation of the obtained results and possible reasons for them. Regarding the Conclusion section, likewise the other parts, it is well written, but in my opinion, maybe a bit too long. Therefore, consider shortening it. There are a few additional comments below:
The abstract seems to be composed of random sentences without a clear flow. Please revise it to adapt the text to interested readers and to make it easier to follow.
Regarding the choice of keywords, polylactic acid, by-product revalorization, bioadhesive, rosehip, mechanical properties, circular economy, I suggest including cassava starch as other components of trays are mentioned, and removing the mechanical properties, as it isn't the only property determined in this study.
The prepared manuscript does not follow the Instructions for Authors for the Foods journal. The type of article is missing, and the order of authors' names and surnames is inadequate.
In the Materials and Methods, include a brief explanation of the formulation of citric acid-modified cassava starch-based adhesives.
Line 227, high proportion of lignin (approximately 44.3 %), from Table 1, it can be seen that lignin proportion was 28.21 ± 0.6 %. Please revise it.
Why FTIR results for thermo pressing at 120°C are missing?
Author Response
We have done the revision of our manuscript following the Reviewers` suggestions. We would like to thank the Reviewers for their appropriate and useful comments and suggestions that help us to improve the manuscript quality. Accordingly, the suggested changes were introduced in the revised version in blue, in order to highlight them.
Comment 1: The abstract seems to be composed of random sentences without a clear flow. Please revise it to adapt the text to interested readers and to make it easier to follow.
Answer: We appreciate the Reviewer’s comments. The abstract has been improved following your suggestions.
Comment 2: Regarding the choice of keywords, polylactic acid, by-product revalorization, bioadhesive, rosehip, mechanical properties, circular economy, I suggest including cassava starch as other components of trays are mentioned, and removing the mechanical properties, as it isn't the only property determined in this study.
Answer: We appreciate the Reviewer’s comments. The keywords were modified following your suggestions.
Comment 3: The prepared manuscript does not follow the Instructions for Authors for the Foods journal. The type of article is missing, and the order of authors' names and surnames is inadequate.
Answer: We appreciate the Reviewer’s comments. Done.
Comment 4 : In the Materials and Methods, include a brief explanation of the formulation of citric acid-modified cassava starch-based adhesives.
Answer: The formulation of the adhesives was detailed in the cited previous works. For this reason, and given that other reviewers suggested shortening this section, we prefer to retain the citation.
Comment 5: Line 227, high proportion of lignin (approximately 44.3 %), from Table 1, it can be seen that lignin proportion was 28.21 ± 0.6 %. Please revise it.
Answer: Thank you, this means that 44,3% of the fiber is lignin, not from the total raw material weight. This was clarified in the text
Comment 6: Why FTIR results for thermo pressing at 120°C are missing?
Answer: Following the Reviewer’s suggestion FTIR results for thermo pressing at 120°C was included as Figure 3b. Likewise a comment about this Figure was included in the revised version.

Round 2
Reviewer 1 Report
Comments and Suggestions for Authors
Dear Authors,
I do not have any comment about your manuscript.
The reviewer
Author Response
We have done the revision of our manuscript following the Reviewers suggestions. We would like to thank the Reviewer for her/his appropriate and useful comments and suggestions that help us to improve the manuscript quality. Accordingly, the suggested changes were introduced in the first revised version in blue while those corresponding to the second revision were highlighted in pink.
Comment 1. The particle size of the rosehip residue (<500 µm) was mentioned, but there was no discussion on how particle size affects the composite properties with homogeneity / performance.
Answer: The maximum particle size was selected to guarantee that all particles within the laminate could be adequately embedded within the polymer matrix and reduce possible roughness.
Comment 2. The mechanical testing focused on puncture resistance. While, other mechanical properties like tensile strength or flexibility were not discussed, which were also important for packaging materials. Please give deep discussion.
Answer: We selected puncture testing as a comparative measure of mechanical properties considering the resources available. Further mechanical testing of the trays are planned for a later phase of the work, in which the trays are to be produced in pilot scale processing lines and their performance in the packaging of different food matrices is evaluated. This step will be oriented to simulate real application conditions, considering variables such as the type of food, and storage time and mechanical requirements of the handling process. The objective will be to determine the functional suitability of the developed materials in representative contexts of use adapted to each need. This information was included in the manuscript.
These statements were included in the revised version and highlighted in pink.
Comment 3. The authors did not compare their material performance with conventional polystyrene (PS) trays beyond density including a direct comparison in terms of cost, biodegradation rate, and lifecycle analysis.
Answer: Expanded polystyrene trays are generally cheaper per unit compared to biodegradable alternatives. However, the additional costs can be offset in the long run by environmental benefits. In contrast, biopolymers such as polylactic acid (PLA) and starch can degrade under composting conditions.
Comparative life cycle analysis studies have shown that biodegradable materials have lower greenhouse gas emissions and lower consumption of fossil resources compared to polystyrene. For instance, Razza et al. 2015 have demonstrated that starch-based materials require 50% less non-renewable energy during production and generate 60% fewer greenhouse gas emissions than conventional expanded polystyrene (PS) containers. Additionally, biodegradable polymers offer a sustainable solution for applications like food containers, where recycling is challenging or impractical, thereby helping to reduce solid plastic waste generation.
These statements were included in the revised version and highlighted in pink. Accordingly, the reference was added.
Cita: 3. Razza, F.; Degli Innocenti, F.; Dobon, A.; Aliaga, C.; Sanchez, C.; Hortal, M. Environmental profile of a bio-based and biodegradable foamed packaging prototype in comparison with the current benchmark. Journal of Cleaner Production 2015, 102, 493–500. https://doi.org/10.1016/J.JCLEPRO.2015.04.033
Comment 4. Why did higher temperatures improve cross-linking? But the authors did not explain why longer processing times at 130°C and did not further enhance properties. This inconsistency should be addressed. Please, explain.
Answer: Citric acid (CA) is one of the most common and green poly(carboxylic acids) and has been utilized in a wide range of chemical applications, including cross-linking (Ciriminna et al. 2017, Isha et al. 2023, Uranga et al. 2020, Monory et al. 2023, Zhang et al. 2023,).
The possible cross-linking of CA with cellulose starts with the dehydration of CA to form a cyclic anhydride followed by esterification with the hydroxyl groups of the cellulose//hemicellulose or starch. At temperatures higher than 120ºC, the CA partially dehydrates to form citric anhydrides, which are even more reactive toward hydroxyl groups (Ma et al., 2021; Uranga et al. 2020;Zhang et al. 2023). These anhydrides readily form ester linkages with polysaccharides, increasing the efficiency of the cross-linking process during thermal treatment. According to Reddy and Yang (2010), time is one of the most important factors determining the efficiency of carboxylic acid cross-linking and hence the properties of the cross-linked materials. Sufficient curing time is necessary for the cross-linking reaction to occur but excess curing will damage the starch molecules.
As regards the effect of processing time at 130°C, it can be observed that thinner materials were obtained at higher times.
These statements were included in the revised version and highlighted in pink. Accordingly, the references were added.
--Ciriminna, R.; Meneguzzo, F.; Delisi, R.; Pagliaro, M. Citric acid: emerging applications of key biotechnology industrial products. Chemistry Central Journal 2017, 11, 22. https://doi.org/10.1186/s13065-017-0251-y
-Isha, D.; Ramandeep, K.M.; Arashdeep, S.; Jaswinder, K. Citric acid: An ecofriendly cross-linker for the production of functional biopolymeric materials. Sustainable Chemistry and Pharmacy 2023, 36, 101307. https://doi.org/10.1016/j.scp.2023.101307
Ma, Y., You, X., Rissanen, M., Schlapp-Hackl, I., & Sixta, H. Sustainable cross-linking of man-made cellulosic fibers with poly (carboxylic acids) for fibrillation control. ACS Sustainable Chemistry & Engineering, 2021, 9(49), 16749-16756.
- Uranga, J.; Nguyen, B.T.; Si, T.T.; Guerrero, P.; de la Caba, K. The Effect of Cross-Linking with Citric Acid on the Properties of Agar/Fish Gelatin Films. Polymers 2020, 12, 291. https://doi.org/10.3390/polym12020291
- Monroy, Y.; Rivero, S.; García, M.A. Liquid and Pressure-Sensitive Adhesives Based on Cassava Starch and Gelatin Capsule Residue: Green Alternatives for the Packaging Industry. Foods 2023, 12, 3982. https://doi.org/10.3390/foods12213982
-Zhang, W.; Roy, S.; Assadpour, E.; Cong, X.; Jafari, S.M. Cross-linked biopolymeric films by citric acid for food packaging and preservation. Advances in Colloid and Interface Science 2023, 314, 102886. https://doi.org/10.1016/j.cis.2023.102886.
Reddy, N; Yang, Y. (2010) Citric acid cross-linking of starch films. Food Chemistry, 118, 702.
Comment 5. The 30% CA/starch ratio was not optimized. There was raising questions about excess acid impacted on material brittleness or toxicity.
Answer: The formulation of the bioadhesives was developed as part of Dr. Monroy's doctoral thesis, and its optimization has been published in previous works cited in the manuscript. These works detail the optimization of cassava starch content, the type of polycarboxylic acid used for its modification, and its concentration. These papers were properly cited in the manuscript.
In addition, CA is a nature-based organic chemical that is usually produced by the fermentation of citrus and is reported as nontoxic within several major regulatory frameworks. CA is classified as GRAS (Generally Recognized as Safe) status in the US (59 FR 63894 and 80N-0218), it is authorized in the EU as a food additive according to regulation N0. 1333/20008, and has been listed in Codex General Standard for Food Additives (INS 330). Nonetheless, specific migration assessment must be conducted prior to any use on food products.
Comment 6. Clarify why PLA+CP+PLA trays had lower contact angles than PLA+L+PLA despite similar processing. Give more deep discussion.
Answer: The FTIR spectra of PLA+CP+PLA have a peak at 2850- 2920 cm-1 with a higher intensity than that of the PLA+L+PLA samples, which may be due to the cellulose fibers being more exposed in these materials, since processing everything together results in greater penetration of the PLA film and exposure of the fibers. As can also be seen in the SEM. Added to the lack of crosslinking that may occur with CA due to the temperature profile that was explained earlier.
Considering this, a deeper discussion was included in the text that now read: “Moreover, both PLA film and PLA+L+PLA trays showed higher average values of 92 ± 3°, with no significant differences with the composite L, all corresponding to hydrophobic surfaces. Nevertheless, an increase in the wettability of the materials was observed in the PLA+CP+PLA system, presenting an average contact angle value of 82 ± 3°. This phenomenon may be attributed to specific processing parameters that potentially altered the core temperature profile of the composite matrix during thermoforming operations, thereby potentially impeding citric acid cross-linkage mechanisms and consequently yielding a material with enhanced hydrophilicity, as was evidenced in laminated structures subjected to reduced thermal conditions (Figure 4). Besides, the in-situ crosslinking of the bioadhesive in the CP within the PLA films led to a greater intercalation of PLA within the composite matrix observed by SEM (Figure 6b). Nonetheless, these could lead to some exposure of the cellulosic and more hydrophilic fillers fibers, leading to an increase in wettability and water resistance (Table 3). The superficial exposure of these fibers was identified with the increase in the FTIR signal at around 2850 and 2920 cm−1 ascribed to symmetric and asymmetric C–H stretching vibrations in lignin, cellulose, or hemicellulose components (Figure 7).”

Reviewer 2 Report
Comments and Suggestions for Authors
The author has addressed all the previous comments, and the revised manuscript looks good. I have no further suggestions or concerns.